# Resorbable Patient-Specific Implants of Molybdenum for Pediatric Craniofacial Surgery—Proof of Concept in an In Vivo Pilot Study

**DOI:** 10.3390/jfb15050118

**Published:** 2024-04-29

**Authors:** Dominik Thomas Hoppe, André Toschka, Nadia Karnatz, Henriette Louise Moellmann, Maximilian Seidl, Lutz van Meenen, Georg Poehle, Christian Redlich, Majeed Rana

**Affiliations:** 1Department of Oral, Maxillofacial and Facial Plastic Surgery, University Hospital Düsseldorf, 40225 Düsseldorf, Germany; dominik.hoppe@med.uni-duesseldorf.de (D.T.H.); andre.toschka@med.uni-duesseldorf.de (A.T.); nadia.karnatz@med.uni-duesseldorf.de (N.K.); henriettelouise.moellmann@med.uni-duesseldorf.de (H.L.M.); 2Institute of Pathology, University Hospital Düsseldorf, 40225 Düsseldorf, Germany; maximilian.seidl@med.uni-duesseldorf.de; 3Karl Leibinger Medizintechnik GmbH & Co. KG, 78570 Mühlheim, Germany; lutz.vanmeenen@klsmartin.com; 4Fraunhofer Institute for Manufacturing Technology and Advanced Materials IFAM, Branch Lab Dresden, 01277 Dresden, Germany; georg.poehle@ifam-dd.fraunhofer.de (G.P.); christian.redlich@ifam-dd.fraunhofer.de (C.R.)

**Keywords:** resorbable, metallic, molybdenum, implant, pediatrics, craniosynostosis, cranioplasty, biocompatible, patient-specific, oral and maxillofacial surgery

## Abstract

Titanium continues to be the gold standard in the field of osteosynthesis materials. This also applies to pediatric craniofacial surgery. Various resorbable materials have already been developed in order to avoid costly and risky second operations to remove metal in children. However, none of these resorbable materials have been able to completely replace the previous gold standard, titanium, in a satisfactory manner. This has led to the need for a new resorbable osteosynthesis material that fulfills the requirements for biocompatibility, stability, and uniform resorption. In our previous in vitro and in vivo work, we were able to show that molybdenum fulfills these requirements. To further confirm these results, we conducted a proof of concept in four domestic pigs, each of which was implanted with a resorbable molybdenum implant. The animals were then examined daily for local inflammatory parameters. After 54 days, the animals were euthanized with subsequent computer tomography imaging. We also removed the implants together with the surrounding tissue and parts of the spleen, liver, and kidney for histopathological evaluation. The molybdenum implants were also analyzed metallographically and using scanning electron microscopy. A blood sample was taken pre- and post-operatively. None of the animals showed clinical signs of inflammation over the entire test period. Histopathologically, good tissue compatibility was found. Early signs of degradation were observed after 54 days, which were not sufficient for major resorption. Resorption is expected with longer in situ residence times based on results of similar earlier investigations.

## 1. Introduction

The term craniosynostosis describes premature pathological ossification of the connective tissue growth plates in the region of the skull bones, which can lead to abnormal skull growth and thus to skull deformity in the child. If ossification is pathologically too early, growth perpendicular to the affected cranial sutures is inhibited and compensatory growth parallel to them is increased. As a result, typical deformities of the skull occur. The etiology of craniosynostoses has not yet been conclusively determined. In general, craniosynostoses are a rather rare clinical picture. The incidence is currently 3 to 6 cases per 10,000 births [1]. Scaphocephalus, in which there is premature closure of the sagittal sutura, is the most common form, with an incidence of 1:2100 to 1:2500 births [2]. The classification of cranial suture synostoses results from the affected suture and the associated typical skull shape. Clinically superficial refers to deformation of the cranium as well as an increase in intracranial pressure due to inhibited cranial growth [3]. The indication for surgical therapy results from functional and aesthetic reasons. Most cases are associated with increasing intracranial pressure and its influence on the brain and visual acuity of the child. The aim of surgical interventions is to reduce intracranial pressure and to remodel the cranial bones [4]. Most surgical procedures are performed between 9 to 12 months of age [5]. In complex operations, parts of the skull are reshaped, manually placed, and fixed by means of titanium implants [6,7]. Conventionally, an open approach is often chosen, although minimally invasive approaches using endoscopes have already been described [8,9]. However, methods of open surgical procedures prove to be very subjective, as the remodeling of the skull is done according to the surgeon’s assessments. This leads to variable, surgeon-specific outcomes with extended operating times. To improve pre-operative planning and reduce morbidity and operative risks, the use of computer-aided design (CAD), manufacturing (CAM), and virtual surgical planning (VSP) is becoming increasingly common [10,11,12].

Despite technical and surgical advancements, titanium implants remain the gold standard for osteosynthesis in oral and maxillofacial surgery [13]. The widespread use of titanium implants is based on their excellent mechanical properties, easy handling, adequate stability, and good biocompatibility. The disadvantages of using titanium implants include the palpability of the implants under the skin and possible discomfort in the area of the implants depending on the ambient temperature [14,15]. In addition, growth restrictions may occur if body growth is not fully completed. As a result, titanium implants are removed in a second operation in 5–38% of cases [13]. In pediatric surgery, in particular, the question of removing the inserted osteosynthesis materials frequently arises. Due to fears of growth inhibition of the affected bones, removal of the inserted osteosynthesis systems after healing is recommended in the literature [16,17].

Against this background, the question arises of a new resorbable osteosynthesis material that resembles the requirements of titanium and at the same time eliminates the need for risky second metal removal procedures in children [18]. Various resorbable materials have already been developed for this purpose [19,20]. These can be divided into two large groups, which are metallic and non-metallic materials.

Magnesium or magnesium alloy-based implants are well investigated and have received market authorization for some indications. Successful applications have been described in orthopedics in particular (e.g., MAGNEZIX^®^ CS from Syntellix AG, Hanover, Germany) [21]. The advantages of the use of magnesium-based implants lie in their good biocompatibility and bone integration [22,23,24]. Despite promising applications, significant drawbacks also emerge. These include the need for greater dimensioning of the implants to achieve the same stability as titanium, as well as unpredictable and irregular resorption [25]. Another group of resorbable osteosynthesis materials are implants made of polylactic acids (PLA) or polyglycolic acid (PGA). These are already being used extensively in oral and maxillofacial surgery, for example, in the systems of Lactosorb^®^ (Biomet Inc., Jacksonville, FL, USA) and RapidSorb^®^ (DePuy Synthes CMF, West Chester, PA, USA) [26]. Another example is the resorbable polymer Poly(D,L-lactide) (PDLLA), which is already well-established in craniofacial and maxillofacial surgery. One example is the SonicWeld Rx^®^ system (KLS Martin, Tuttlingen, Germany). However, low mechanical load-bearing capacity with consequently larger dimensions and post-operative complications such as inflammation, immune reactions, and wound dehiscence are problematic in application [27].

In summary, there are currently no resorbable materials that can be compared to gold-standard titanium. Against this background, molybdenum (Mo) represents an interesting alternative to previous materials [28]. As an essential trace element, it is found in the human body and is excreted through the kidneys [29]. At the same time, it shows sufficient mechanical stability and biocompatibility. In a series of in vivo experiments by Schauer et al., it was shown that molybdenum arches (ø 250 µm) implanted in the abdominal aorta of Wistar rats had a degradation rate of 13.5 µm/year [30]. In a similar series of experiments by Sikora-Jasinska et al., it was shown that six months after implantation of molybdenum stents in the artery of mice, considerable resorption occurred. The original diameter of the inserted molybdenum implant was 128.9 ± 2.4 μm. After six months of in vivo storage, a reduction of 34.5 ± 1.5% was observed [31].

In our recent work on molybdenum, it was demonstrated through in vitro and in vivo experiments that molybdenum has excellent biocompatibility, stability, and uniform absorption [32]. For this purpose, we implanted molybdenum and titanium implants subcutaneously in the nuchal fold of 96 rats. Subsequently, we removed the implants, along with the spleen, liver, and kidney from one-third of the test animals after 12, 24, and 52 weeks. After histopathological evaluation and metallographic assessment, good biocompatibility was demonstrated. However, the resorption of the implants was below the expected resorption. This is most likely due to the encapsulation of the implants, which was subsequently revealed in the histopathological evaluation [32].

Following Krenn et al., we analyzed the surrounding tissue of the implants for capsule formation according to the Krenn score. After 12 weeks, some of the test animals already showed capsule formation around the implants, which is retrospectively seen as a reason for a lack of resorption of the molybdenum implants [32]. For the test times for removing the implants, we orientated ourselves on the in vitro test we had previously carried out. In this test, biocompatibility was demonstrated on cells in accordance with ISO standard 10993-5 [33]. We also investigated the mechanical strength of pure molybdenum sheet metal of 95 μm thickness. For this purpose, we stored the test pieces in a simulated body fluid (SBF) according to Kokubo with a pH value of 7.4. After 2, 4, and 6 months, we tested the tensile strength of the molybdenum pieces. After 6 months, there was a loss of mechanical strength of about 50%. Based on the data, a complete loss of stability was expected after 7.5 to 9 months. [34].

Based on the positive results, the next step and goal of the experiment was the fabrication of a generic implant model suitable for semi-automated fitting for specific geometries. In addition to the applicability of the new resorbable implants made of molybdenum, biological compatibility, stability, and osteointegration were to be assessed. In addition, the inserted molybdenum implants should be examined for possible resorption. Our preliminary work in the rat model already showed capsule formation after 12 weeks. One of the aims of this pilot study was to investigate whether such capsule formation also occurs under direct contact between the molybdenum implant and bone at this early stage and prevents possible resorption. We therefore removed the molybdenum implants after 54 days.

## 2. Materials and Methods

### 2.1. Materials

#### 2.1.1. Animal Model

Four domestic pigs at 10–14 percent of their adult weight (250 kg) were included in the experiment. Accordingly, the animals had a weight of 24–34 kg. The animals came from the Ferkelerzeugergemeinschaft Springe-Burgdorf e.V. (Niedersachsenstraße 19, 31832 Springe, Deister, Germany). Two female and two male pigs were selected to control for sex-related results. The pigs were initially housed in a separate area with no direct contact with other animals. Specific protective clothing was worn and regular cleaning and disinfection of the working paths, stall, and functional areas were carried out. The pigs were fed with commercially available pig feed throughout the entire period. There was always free access to drinking water. Prior to surgery, the pigs were allowed to fast for a maximum of 18 h and continued to have access to water. In the chronic part of the experiment, animals were kept post-operatively in a recovery box (2.5 × 2.5 m) on a rubber mat under a heat lamp until full awakening. On the first post-operative day, re-housing in small groups of no more than four animals was provided.

All trials were approved by the Lower Saxony State Office for Consumer Protection and Food Safety (LAVES Niedersachsen, Germany; file number 33.19-42502-04-22-00178), which is responsible for processing animal experiment applications in accordance with the Animal Welfare Act (TierSchG) in Lower Saxony.

#### 2.1.2. Implants

The implants were manufactured by KLS Martin (Tuttlingen, Germany) using the selective laser melting process (SLM) and prepared for surgical use. Specimens were produced on SLM 125 machines (SLM Solutions, Lübeck, Germany), which follow an identical build strategy. In this process, a thin layer of powder is applied to a building platform, and subsequently, a laser is used to melt the powder particles together. The building platform is then lowered, and a new layer of powder is applied and fused. This layer-by-layer approach enables the construction of the desired implant geometry. Each specimen was designed with multiple hexagon structures (Figure 1), a ø 14 mm diameter, and a bar dimension of 3.5 × 1.3 mm (width and thickness). The overall dimension of the structure should cover a defect size of 25 × 40 mm; therefore, an outline contour of 40 × 50 mm was chosen. KLS Martin utilized molybdenum powder (Mo-PDMPB) obtained from H.C.—Starck (Goslar, Germany), which had a molybdenum content of >99.5% and a particle size between 15–44 μm. To achieve the desired surface finish, the specimens underwent plasma polishing for 7 min using a nontoxic electrolyte developed and applied by H&E (Moritzburg, Germany), resulting in an average surface roughness of Rz max < 30 μm. To ensure the safety and suitability of the implants for surgical applications, the fabricated specimens underwent gamma sterilization at a high dose (≥25 kGy). For specimen fixation, pins with a diameter of 1.1 mm and a length of 4.9 mm were produced using identical SLM parameters and raw materials.

### 2.2. Methods

#### 2.2.1. Study Design

A total of four animals were implanted with an implant made of molybdenum. Pre-operatively, a blood sample was taken. Subsequently, the animals were visited three times a day for the first three post-operative days. Thereafter, two visits were made daily until the end of the experiment. The evaluation was based on the Pig Grimace Scale according to the recommendations of the Society of Laboratory Animal Science (GV Solas) [35]. In addition, possible local signs of inflammation were documented. After 4 weeks, a blood sample was taken from all test animals again. The animals were then euthanized. Immediately afterwards, computer topographic imaging of the animals’ skulls and removal of the spleen, liver, and kidney took place (Figure 2). In addition, the implants were removed together with the surrounding tissue. Finally, histopathological evaluation of the tissue samples and metallographic analysis of the retrieved implants were performed.

#### 2.2.2. Pre-Operative Management and Premedication

All four domestic pigs were operated on under general anesthesia with tracheal intubation. Food restriction was initiated 18 h before premedication, and water access was maintained until just before premedication. Premedication with 2 mg/kg weight-adapted azaperone (Stresnil^®^ Elanco GmbH, Cuxhaven, Germany) and 10–15 mg/kg weight-adapted ketamine (Ketaset^®^ Zoetis, Parsippany, NJ, USA) was performed in the preparation room. Venous access via an indwelling peripheral venous catheter (PVK) was obtained through the ear vein. For this purpose, this area was shaved, cleaned, and disinfected. Propofol (Disoprivan^®^ Aspen Germany GmbH, Munich, Germany) at a 3 mg/kg weight-adapted dose was then applied. Intubation was performed by means of an endotracheal tube. Eye ointment (Bepanthen^®^ Bayer, Grenzach-Wyhlen, Germany) was applied to the conjunctival sacs to protect the eyes. The animal’s skin was cleared of bristles and cleaned while still in the preparation room in the appropriate surgical area.

#### 2.2.3. Surgical Implantation of the Molybdenum Implants

In the operating room, the animal was connected to the anesthesia machine. Inhalation anesthesia was induced by a mixture of the anesthetic gas isoflurane (Sedaconda^®^ Sedana Medical AB, Danderyd, Sweden) and oxygen.

The respiratory rate was set at 14–16/min and the respiratory volume at 8–10 mL/kg body weight.

Intraoperative analgesia with fentanyl (bolus 0.002 mg/kg and continuous drip 0.003–0.005 mg/kg/h) was administered. An isotonic electrolyte solution was administered for volume substitution. The amount was 5–10 mL/kg/h.

We placed the animals in a prone position on a warming mat with the limbs restrained to stabilize the position. Then, we cleaned and disinfected all the places where a skin incision or puncture was made (Figure 3a,b). For local anesthesia, we injected the local anesthetic mepivacaine (Meaverin^®^ PUREN Pharma GmbH & Co. KG, Munich, Germany) into the incision margins before each incision. First, we detached the skin galea flap from the periosteum approximately up to the supraorbital bulge. Laterally, the flap was dissected over the temporal fascia. We were then able to cut and detach the periosteum along the coronal sutures and along the attachment of the temporalis muscle. It was left frontally pedicled to be used to cover defects. After preparation, the osteotomy lines were drawn. These described a rectangle of 2.5 × 5 cm and included parts of the sagittal sutura. The next step was the removal of a bone flap, for which the 2.5 × 5 cm rectangle was sawn out with an oscillating saw (Figure 3c). The bone flap was then lifted with a raspatory. We left the internal lamina intact and avoided exposing the dura mater (Figure 3d). We were then able to bridge the resulting lesions by inserting the new molybdenum implants (Figure 3e). Afterwards, the inserted implants could be fixed with the help of prefabricated pins. To do this, we drilled holes in the bone in the area of the implant edges, into which the pins could then be inserted with clamping tension. We were able to use the drill holes already made in the implants as a guide. After fixing the implants, the periosteal flap was folded back, covering most of the bone gap. After repositioning the skin–galea flap, we performed a two-layer wound closure. For this, we used Vicryl sutures (Vicryl^®^ Ethicon, Raritan, NJ, USA) to avoid suture removal (Figure 3f). Finally, local anesthesia was repeated with bupivacaine (Carbostesin^®^ AstraZeneca PLC Cambridge, UK).

Vital signs were monitored regularly by electrocardiogram, pulse oximeter, reflex testing, and blood pressure measurement, and were documented in the anesthesia record every 20 min to verify adequate depth of anesthesia as well as analgesia.

#### 2.2.4. Post-Operative Management

At the end of the surgical procedure, the surgical wounds were disinfected and cleaned. Finally, a spray bandage was applied. The anesthesia was terminated after safe stabilization and extubation followed. The animal was given analgesic with butorphanol (Dolorex^®^ Rahway, Rahway, NJ, USA) at a dose of 0.2 mg/kg weight-adapted. The animal was then transported to the stable area where its vital signs were checked. Here, respiration was monitored for one hour based on oxygen saturation and the animal was observed clinically. The animals were then visited daily for 54 days and checked for local signs of inflammation. Three visits were made daily during the first post-operative week. In addition, pain medication with 4 mg/kg carprofen (Rimadyl^®^ Zoetis Deutschland GmbH, Berlin, Germany) and 0.2 mg/kg weight-adapted butorphanol (Dolorex^®^ Rahway, Rahway, NJ, USA) was given to all animals for five days post-operatively. An antibiotic shield with amoxicillin/clavulanic acid was also given for the above-mentioned period.

#### 2.2.5. Euthanasia, Computer Tomography, and removal of Implants and Organs

After 54 days, the pigs underwent a final clinical examination and were then euthanized. The animals were euthanized under anesthesia. Premedication was carried out with xylazine (Rompun^®^ Bayer Animal Health GmbH, Leverkusen, Germany) and induction of anesthesia with propofol (Disoprivan^®^ Aspen Germany GmbH, Munich, Germany). Euthanasia was then performed under deep anesthesia with potassium chloride (1–2 mmol/kg, equivalent to 75–120 mg/kg). Afterwards, computer tomography of the four pig skulls was performed (Figure 4a,d,e,f). After imaging, the spleen, liver, and kidney were removed from all four animals for subsequent histopathological evaluation. In addition to the organ samples, we explanted the inserted implants, including surrounding hard and soft tissue in all four animals (Figure 4b,c). We also performed a histopathological examination of these tissues and a metallographic evaluation of the removed implants for material properties.

#### 2.2.6. Histopathological Processing and Examination

Two bone samples were taken from each animal for further embedding and analysis. The tissues were then fixed in formaldehyde. For further processing, the molybdenum implants were removed from the skull bones and the bone samples were decalcified using ethylenediaminetetraacetic acid (EDTA). The extracted molybdenum implants were sent to the Fraunhofer Institute for Manufacturing Technology and Advanced Materials IFAM Dresden (Germany) for further metallographic analysis. After paraffin embedding, the blocks were cut into 1.5 μm thick layers. In addition, H&E staining was performed with subsequent digitization of the slides. We used the Aperio slide scanner (Leica Biosystems, Nussloch, Germany) with a 40× objective lens and a resolution of 0.2529 μm per pixel. For further analysis and evaluation of the digital slides, we used the QuPath software (PMID: 29203879), version 0.4.4. The preparation and evaluation of the tissue samples was carried out in collaboration with the histopathological institute of the University Clinic Düsseldorf (Düsseldorf, Germany). The cranial bone specimens were analyzed in accordance with ISO 10993-6 for the biological evaluation of medical devices. The region of interest (RIO) for the evaluation was the bone tissue in the area around the site of implant placement and, in particular, the area of direct contact between the molybdenum implant and the newly formed bone beneath it. In addition, the defect sites were analyzed for the strength of the newly formed bone in the area of the previously surgically created defect situation.

#### 2.2.7. Analysis of the Metallographic Cross Sections

For metallographic analysis, the molybdenum implants were first removed from the skull bone. After cleaning, the metallographic cross-sections of the explanted metal implants were prepared by embedding in resin and then by metallographic preparation: Embedding in resin served to improve the handling of the samples and to support the edge areas to avoid possible fractures. Metallographic preparation was used to visualize the grain structures of the samples. For this purpose, the embedded implants were first ground with SiC paper in three steps (with successively finer grain size), followed by polishing in two steps with aluminum oxide and diamond suspension. The surface of the polished cross-section was then examined using a light microscope (Zeiss Axiocam 208 color, Oberkochen, Germany). In addition, the surface of an explanted sample in the as-is state (i.e., without metallographic preparation) was analyzed using EDS (energy dispersive X-ray spectroscopy) in an SEM (scanning electron microscope) to determine the elemental composition of possible degradation products. For this purpose, we used the JSM-IT800 device (JEOL GmbH, Germany). This is a field-emission scanning electron microscope that provides ultra-high-resolution images with a resolution limit of up to 1 nm. This means that two image points with a distance of 1 nm can still be distinguished, allowing the microstructure of metals to be analyzed down to the nano range. This also makes it possible to analyze the local chemical composition, crystal structure, and orientation.

## 3. Results

A total of four test animals were included in the trial and operated on. None of the test animals had to be removed from the experiment prematurely. From the first post-operative day until euthanasia, none of the test animals showed any clinical signs of inflammation or intolerance. After just a few days, the cutaneous access routes healed without complications, leaving a dense, irritation-free scar.

The four animals operated on (two female and two male pigs) weighed 29.00 ± 5.00 kg at the time of surgery. The operation took an average of 35.00 ± 27.80 min.

Before the molybdenum implants were inserted, the animals underwent a laboratory blood analysis (Table 1).

The leucocyte count was 12.60 ± 6.02 G/l, the erythrocyte count was 6.03 ± 0.93 T/l, and the platelet count was 121.00 ± 74.10 G/l. The hemoglobin value was 102.00 ± 17.80 g/l and the hematocrit was 0.30 ± 0.05 I/l. The other erythrocyte indices were as follows: MCV 50.20 ± 0.675 fl, MCH 17.20 ± 0.41 pg, MCHC 341.00 ± 5.60 g/l.

Leukocytosis is observed in the test animal 11/23. In the animals with the identification numbers 12/23 and 13/23, anemia was present, which was accompanied by leucopenia in animal 13/23.

Furthermore, a parasitological fecal examination was carried out on animal 13/23 without detecting any pathogens.

After 4 weeks, a further laboratory chemical blood test was carried out on the test animals (Table 2). The post-operative blood count showed 11.50 ± 2.32 G/l leucocytes, 6.05 ± 0.33 T/l erythrocytes, and 331 ± 104 G/l platelets. The post-operative hemoglobin value was 106.00 ± 5.91 g/l and the hematocrit was 0.32 ± 0.01 I/l. The other post-operative erythrocyte indices are as follows: MCV 52.00 ± 0.65 fl, MCH 17.50 ± 0.42 pg, MCHC 335.00 ± 6.35 g/l.

The experimental animals 11/23, 13/23, and 14/23 had anemia post-operatively. Animal 14/23 also showed leukopenia.

Comparing the pre-operative and post-operative values, the following is noticeable. The leucocytes and erythrocytes drop post-operatively, but there is no significant difference with t(3) = 1.04, *p* = 0.373 for the leucocytes and t(3) = 0.208, *p* = 0.848 for the erythrocytes. For the platelets, a significant increase can be observed with t(3) = −3.45, *p* = 0.041 (Figure 5).

For the hemoglobin values, with t(3) = −0.0812, *p* = 0.940, as well as for hematocrit values with t(3) = −0.215), *p* = 0.844, no significant difference can be found (Figure 6).

For the comparison of the erythrocyte indices, only the comparison of the pre- and post-operative MCV with t(3) = −3.53, *p* = 0.039 shows a significant increase. The comparison of MCH with t(3) = −1.25, *p* = 0.299 as well as of MCHC with t(3) = 0.538, *p* = 0.628 is not significant (Figure 7).

Following euthanasia, the four pig skulls underwent computerized tomography to assess the implant position and implant stability, and to rule out possible local osteolysis. None of the test animals showed radiographically visible osteolytic processes. There was no dislocation in any of the implants. None of the inserted molybdenum implants showed a clinically relevant loss of stability such as extensive implant fractures or metal loosening. Only one implant (test animal 11/23) showed a minimal marginal fracture without severe dislocation (Figure 8).

The evaluation of new bone formation in the cranial implants shows an average thickness of 2.34 ± 0.86 mm. Histopathological analysis of the specimens according to the ISO 10993-6 grading scheme revealed 2.63 ± 0.92 polymorphonuclear cells. The other cell types include 3.13 ± 0.99 lymphocytes, 1.13 ± 0.64 macrophages, 0.625 ± 0.74 giant cells, and no plasma cells. Examination of the tissue revealed no necrosis, infection, tissue degeneration, fatty infiltrates, or fibrinous exudates. Half of the samples (n = 4) showed minimal neovascularization. Fibrocytes or fibrous connective tissue showed values of 0.875 ± 0.835. Calculating the total score results in an average value of 3.38 ± 0.52.

One sample showed a cyst measuring 8.84 × 6.10 mm with respiratory epithelium, which is most likely post-traumatic (Figure 9).

The histopathological evaluation of the removed parts of the liver, spleen, and kidney showed no reactive changes or deposits of molybdenum.

After explantation, none of the molybdenum implants showed any macroscopically visible resorption. The metal surface showed a greyish-bluish and inhomogeneous coloration. This indicates some surface degradation in the form of thin oxide films which cause the discoloration.

The metallographic analysis of the removed molybdenum implants confirmed the macroscopic findings. None of the four implants removed showed any resorption yet. However, compared to other investigations of molybdenum implants, the residence time after implantation was much shorter (54 days vs. up to 1 year). The formation of an oxide layer was also not evident. Inside the samples, an inhomogeneous, partly porous, and partially incompletely fused metallic structure was found. These findings were evident in all four implants removed, regardless of the test animal. Due to the undetectable resorption, no difference was apparent compared to the reference sample, which was not used in situ. Overall, the surface appears relatively rough when viewed under a microscope (Figure 10).

The scanning electron microscope (SEM) showed a lighter basic structure (especially the heavier molybdenum) and darker features (the area with less molybdenum) in the BED contrast. The degradation products of molybdenum normally appear lighter in color because they contain oxygen, calcium, sodium, and traces of other elements. The EDX measurements also showed these elements qualitatively in the darker areas. There were therefore detectable decomposition products in the darker areas, albeit in small amounts. The EDX measurements in the lighter areas showed predominantly molybdenum. Apparently, there were few degradation products present (Figure 11).

## 4. Discussion

Following our previous successful in vitro and in vivo experiments on the biocompatibility of molybdenum in the organism [32,34], our overriding aim was to demonstrate the feasibility of using new resorbable implants made of molybdenum as they could later be used in everyday clinical practice. In addition, we wanted to explore the biocompatibility of the new molybdenum implants under direct bone contact in the organism as well as the stability of the implants. Following on from this, we wanted to investigate the resorption behavior under clinical conditions.

It should be mentioned in advance that all the above results are from a pilot study, which is limited by the lack of a comparison group. A direct scientific comparison between different materials is not possible as no groups with already known materials such as titanium or new magnesium implants were used.

We were able to confirm the previously demonstrated biocompatibility in a small cohort. None of the test animals exhibited clinical inflammatory parameters over the entire test period. The comparison of the pre- and post-operative leucocyte count confirms the clinical findings: The leucocytes drop post-operatively, but there is no significant difference with t(3) = 1.04, *p* = 0.373 for the leucocytes. In the post-mortem computed tomography, osteolysis in the area around the implant could be excluded, which is a good radiological indication that molybdenum did not influence the healing of the defect or contribute to inflammatory osteolytic processes.

In the histopathological examination of the organs of the spleen, liver, and kidney, we were able to rule out a deposition of molybdenum in the organs. In particular, the kidneys showed no pathological changes, which is of great importance as molybdenum is mainly excreted renally [36]. However, it should be critically noted that this was a short trial period without significant degradation of the molybdenum implants. Therefore, it cannot be concluded that a longer in situ residence time and more pronounced resorption will not lead to deposits in the organs. It has been described that doses of 80 mg/kg/d can lead to renal dysfunction [36,37]. However, doses of 80 mg/kg/d, equivalent to 5.6 g/d for an adult of 70 kg body weight, are several orders of magnitude larger than the expected release of Mo ions from molybdenum-based implants.

By removing the molybdenum implants together with the surrounding bone tissue and subsequent histopathological evaluation, we were able to show that there was an average new bone formation of 2.34 ± 0.86 mm around the implants. This confirms that molybdenum has no negative impact on defect healing in the tissue, even under direct bone contact. Integration into the bone appears to be an important prerequisite for the subsequent resorption of the implants [38]. Examination of the tissue revealed no necrosis, infection, tissue degeneration, fatty infiltrates, or fibrinous exudates. Only one sample from animal 12/23 showed a cyst measuring (8.84 × 6.10 mm) with respiratory epithelium, which is most likely post-traumatic.

We were able to show in the computed tomography that there was no loss of stability or dislocation of the implants over the entire trial period. The previously created bone defects were reliably bridged by the inserted molybdenum implants. Only in test animal 11/23 did a marginal implant fracture occur. However, this did not appear to have any further influence on the course of the experiment. It is unclear whether the fracture occurred post-operatively as a result of trauma to the animal’s head area during husbandry or whether the implant was weakened during manufacture. It is a known problem that porosity and pore formation can occur when processing pure molybdenum using SLM [39]. It should be noted that open-pored parts in the surface were specifically desired in order to improve the absorption properties. However, the metallography also showed that porosity occurred in the planned dense central implant parts, which was not intended. Therefore, it cannot be ruled out that this could have led to a fracture of the implant. Despite sufficient stability over the test period, it therefore appears necessary to improve the manufacturing process of the molybdenum implants in future to avoid the formation of pores. Porosity should be reduced as much as possible in order to reliably prevent implant fractures and premature loss of stability in everyday clinical practice.

Based on the analyses of the metallographic sections, it should be noted that there was no significant resorption of the molybdenum implants over the entire test period. However, it is positive to note that there were isolated deposits of degradation products in the SEM images. This could be an indication of incipient resorption of the molybdenum implants [40,41,42]. Capsule formation around the implants, as was suspected in our earlier small animal experiment with Wistar rats [32], cannot be the cause of the lack of or low resorption, as complete implant integration with direct bone contact took place. Rather, the short test duration of only 54 days appears to be the cause. It is known from other experiments on the resorption of molybdenum in organisms that resorption of molybdenum in organisms can be expected after a longer trial period. After 1–2 months, the thickness of degradation product layers is at most 1–2 µm even in the case of in vitro degradation testing, where higher degradation rates compared to in vivo studies were observed. Schauer et al. showed that molybdenum wires (ø250 µm) implanted in the abdominal aorta of Wistar rats exhibited a degradation rate of 13.5 µm/year [30]. In a similar series of experiments, Sikora-Jasinska et al. showed that 6 months after the implantation of molybdenum wires in the artery of mice, significant resorption occurred. The original diameter of the molybdenum implant was 128.9 ± 2.4 μm. The area of metallic molybdenum decreased by 34.5 ± 1.5% after six months of in vivo implantation, leaving a wire with a diameter of 85.2 ± 2.6 μm surrounded by a 21.2 ± 1.8 μm thick corrosion film [31]. After 1–2 months, the thickness of degradation product layers is at most 1–2 µm even in the case of in vitro degradation testing reported from earlier studies, where higher degradation rates compared to in vivo studies were observed. It may be the case that degradation rates in bone tissue differ from those observed when molybdenum is surrounded by soft tissue, as was the case in these two studies. In the literature, removal of the osteosynthesis plates in children in the head region is recommended after 3 to 4 months in order to avoid growth restrictions [43]. In our previous in vitro experiment, we were able to demonstrate an expected loss of stability after 7.5–9 months [34]. However, the use of resorbable osteosynthesis implants made of PLA and PGA in the head region in children with a degradation period of 12–18 months had no influence on growth behavior [44], so a degradation period of 7.5 to 9 months until loss of stability appears to be acceptable for molybdenum implants.

A longer in vivo trial of molybdenum implants for osteosynthesis under direct bone contact therefore seems necessary in order to be able to make further statements regarding resorption.

## 5. Conclusions

In summary, it can be said that the exploratory approach made it possible to test molybdenum as a potential resorbable osteosynthesis material under clinical conditions for osteosynthesis and to successfully demonstrate the feasibility of its use. We were able to demonstrate the already known good biocompatibility of the material under direct bone contact. We were also able to show that an unwanted premature loss of stability of the implants does not occur with molybdenum. After 54 days, metallographic cross-sectional imaging still showed no significant resorption. Conclusions about deposits in the organs are therefore hardly possible.

In conclusion, it was shown that molybdenum has very good biocompatibility and that defect healing and bone apposition are possible even under direct bone contact. With regard to the resorption behavior, further in vivo tests over a longer period of time seem necessary.

## Figures and Tables

**Figure 1 jfb-15-00118-f001:**
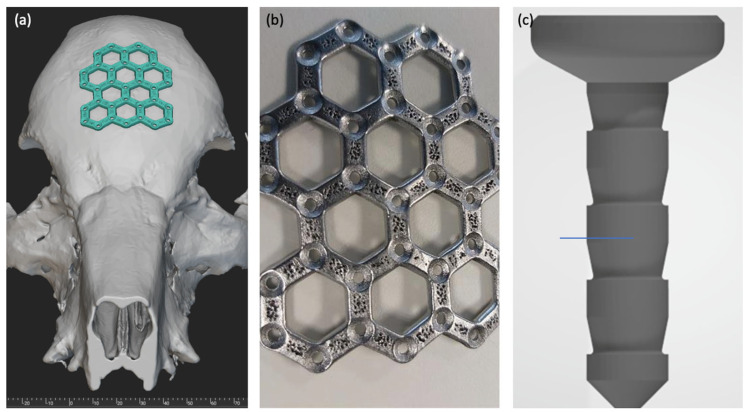
Representation of digital implant planning and a finished implant. (**a**) Digital planning and adaptation of the implant design to the skull’s geometry. (**b**) Industrially manufactured and already plasma-polished implant made of molybdenum before insertion. (**c**) 4.9 mm long pin for individual attachment of the implants.

**Figure 2 jfb-15-00118-f002:**
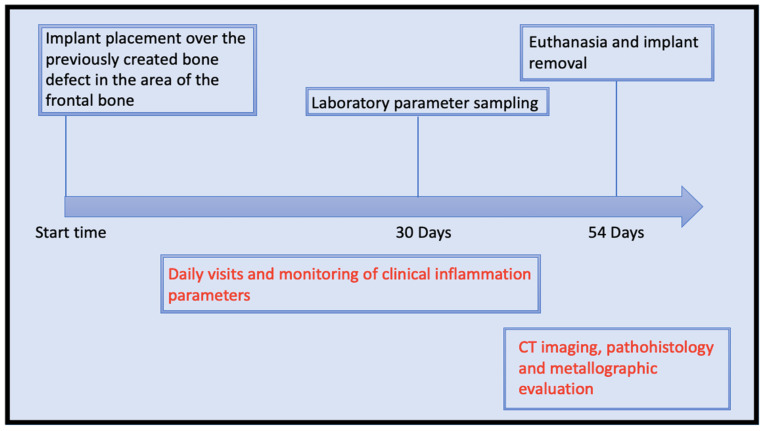
Timeline of the in vivo trial.

**Figure 3 jfb-15-00118-f003:**
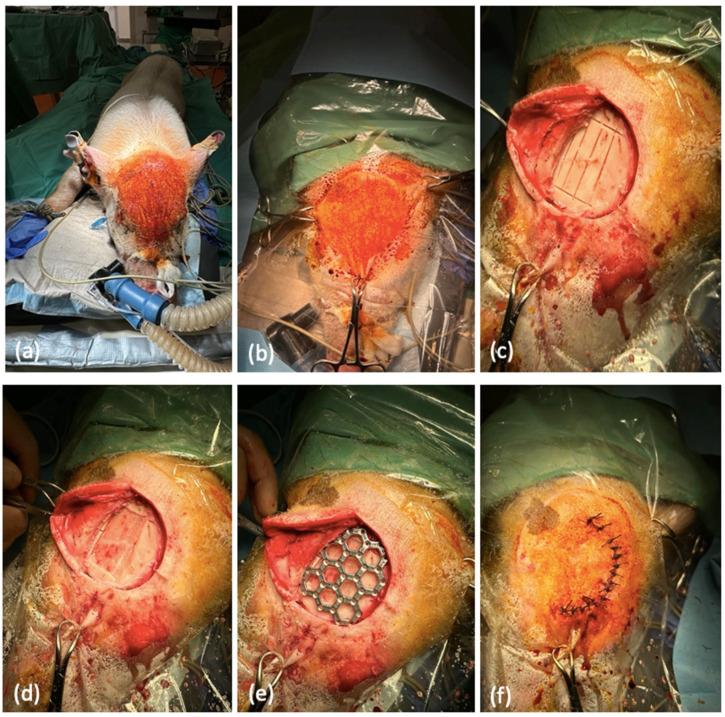
Surgical implantation of the molybdenum implants. (**a**) Positioning and intubation pre-operatively in the operating room with the surgical area already cleaned and disinfected. (**b**) Sterilely covered surgical area in the region of the os frontale. (**c**) Incision of an already severed periosteum in the region of the os frontale. In addition, an already created defect of 2.5 × 5 cm in size. (**d**) Removed externa of the skullcap and view of the interna, which was left completely intact. (**e**) Inserted and fixed molybdenum implant over the defect situation. (**f**) Finally, multi-layer wound closure and suturing using Vicryl sutures.

**Figure 4 jfb-15-00118-f004:**
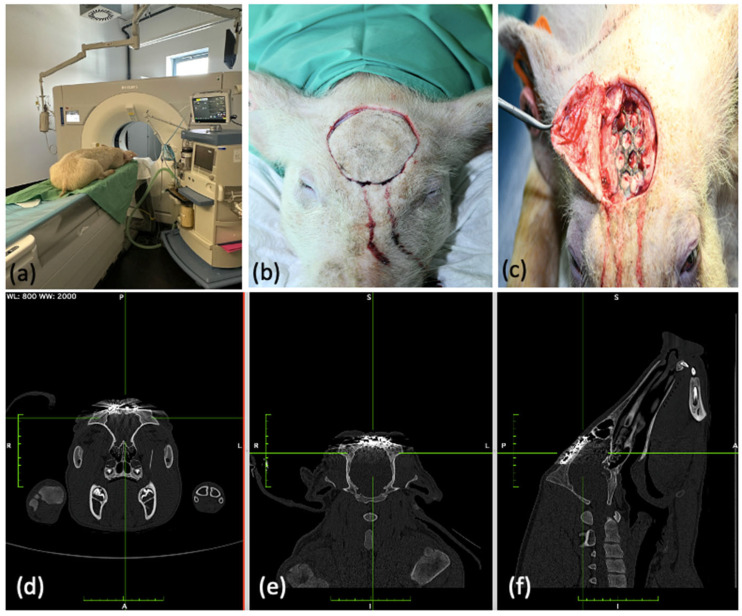
Post-operative computed tomography as well as explantation of the molybdenum implants after euthanasia. (**a**) Computed tomography of the skull using Phillips Iron Spectral CT. (**b**,**c**) Exposure and in situ molybdenum implant with healed defect situation and successful osseointegration. (**d**–**f**) Computed tomographic imaging of the oseointegrated molybdenum implants in coronal, axial, and sagittal planes.

**Figure 5 jfb-15-00118-f005:**
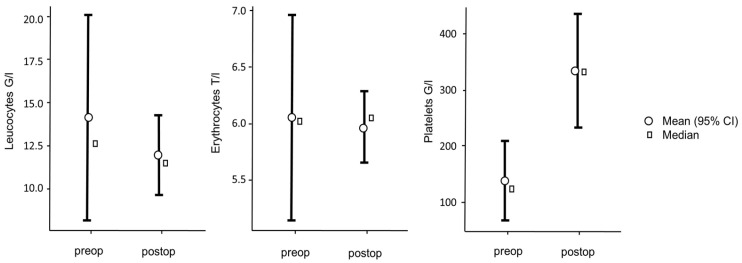
Comparison of the pre- and post-operative blood count.

**Figure 6 jfb-15-00118-f006:**
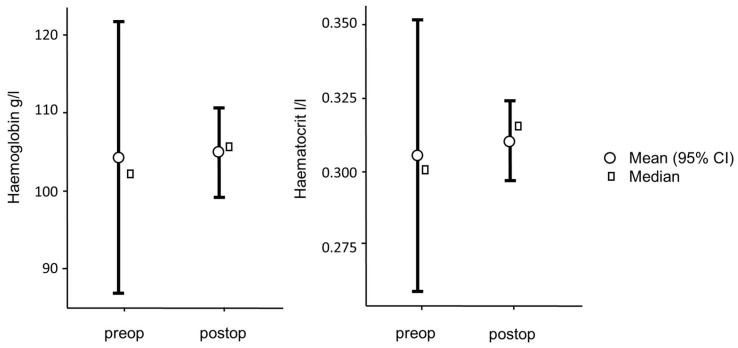
Comparison of the pre- and post-operative hemoglobin and hematocrit.

**Figure 7 jfb-15-00118-f007:**
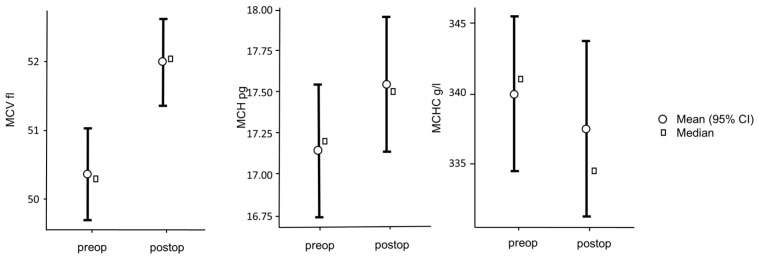
Comparison of the pre- and post-operative erythrocytes-indices.

**Figure 8 jfb-15-00118-f008:**
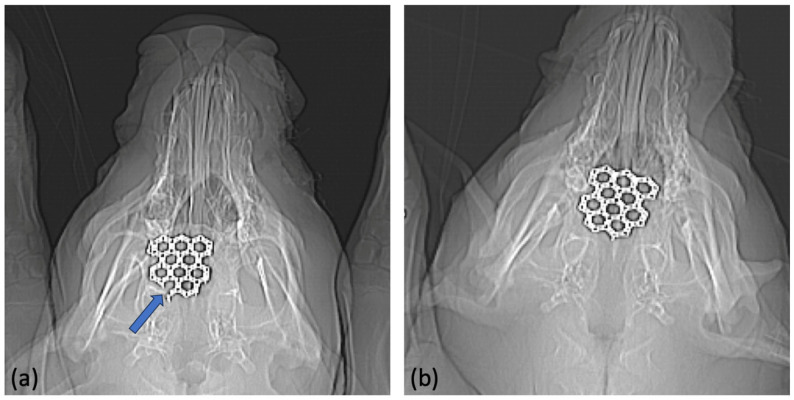
Post-operative X-ray imaging. (**a**) Shows the molybdenum implant in situ (animal 11/23). Minimal fracture of the implant in the marginal area (see arrow). (**b**) Shows an intact molybdenum implant without signs of implant dislocation or implant fractures (animal 13/23).

**Figure 9 jfb-15-00118-f009:**
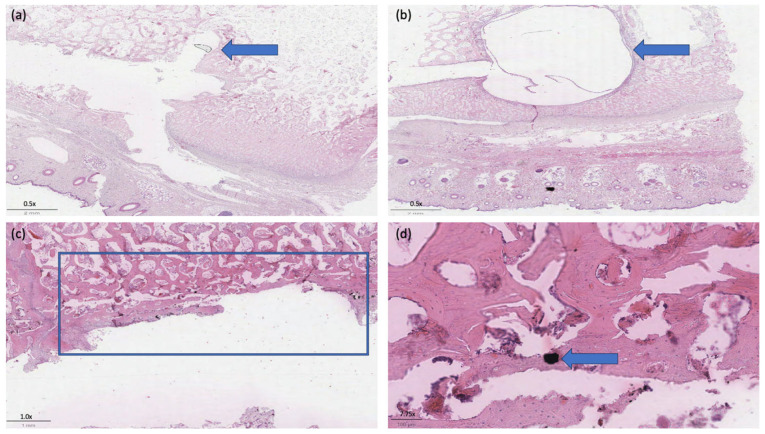
Example of the histopathological evaluation of the tissue surrounding the implants. (**a**) Shows the former position of the molybdenum pin with the surrounding newly formed bone structure (animal 13/23), (see arrow top left). (**b**) Shows the cystic tissue, which is most likely post-traumatic (animal 12/23), (see arrow above right). (**c**) Exemplifies the ROI below the implantation site for evaluation of the surrounding bone structure (see outline), (animal 11/23). (**d**) A remaining molybdenum particle can be seen, which was integrated into the newly formed bone tissue (see arrow bottom right), (animal 14/23). H&E staining was performed.

**Figure 10 jfb-15-00118-f010:**
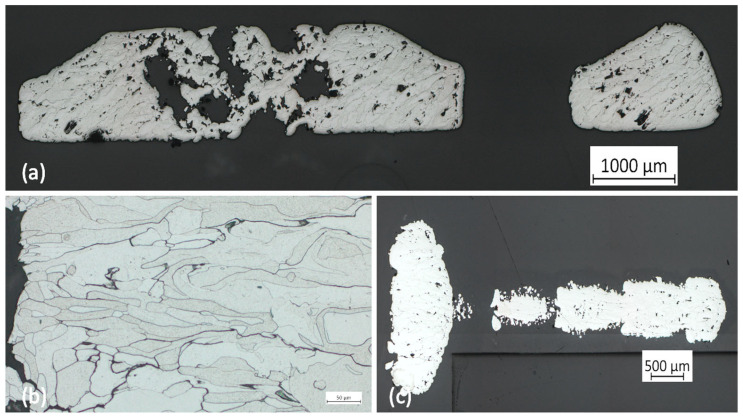
Metallographic cross-sections of the molybdenum implants. (**a**) A cross-section of an implant bar is shown. The porous structures are recognisable. (**b**) The picture shows a section of an implant with an illustration of the partly inhomogeneous metal structure. The incomplete fusion of the molybdenum particles can be seen. (**c**) A fastening pin made of molybdenum is shown in longitudinal section. The cross-section through the pin is slightly off the centre axis, which explains the gaps.

**Figure 11 jfb-15-00118-f011:**
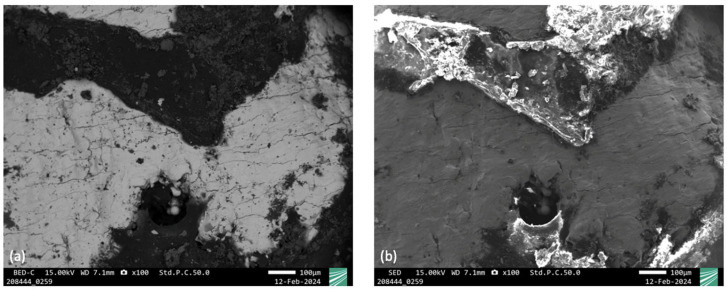
SEM micrograph of the surface of a strut of sample [13/23] after explantation (left: BED contrast; right: SED contrast). (**a**) Shows the SEM micrograph in BED contrast: Darker area in the BED image correspond to detection of O, Ca, and Na by EDX, indicating the formation of a thin layer of degradation products in some areas of the surface. In contrast, in the areas appearing bright, EDX indicates that the formation of degradation products is negligible. (**b**) Shows the SEM micrograph in SED contrast: The image mainly shows surface topographic features such as edges and height differences.

**Table 1 jfb-15-00118-t001:** Overview of pre-operative blood analysis.

Animal Identification	Leucocytes (10–22 G/l)	Erythrocytes (5.8–8.2 T/l)	Haemoglobin (108–148 g/l)	Haematocrit (0.33–0.45 I/l)	MCV (50–65 fl)	MCH (17–21 pg)	MCHC (300–350 g/l)	Thrombocytes (180–600 G/l)
11/23 ♀	(+) 22.70	7.05	124.00	0.36	51.10	17.60	344.00	162.00
12/23 ♂	12.60	5.12	(−) 88.00	(−) 0.26	50.80	17.20	338.00	78.00
13/23 ♀	(−) 8.60	5.42	(−) 90.00	(−) 0.27	49.80	16.60	333.00	79.00
14/23 ♂	12.60	6.63	114.00	0.33	49.80	17.20	345.00	232.00

**Table 2 jfb-15-00118-t002:** Overview of post-operative blood analysis.

Animal Identification	Leucocytes (10–22 G/l)	Erythrocytes (5.8–8.2 T/l)	Haemoglobin (108–148 g/l)	Haematocrit (0.33–0.45 I/l	MCV (50–65 fl)	MCH (17–21 pg)	MCHC (300–350 g/l)	Thrombocytes (180–600 G/l)
11/23 ♀	14.90	5.99	(−) 104.00	0.31	51.80	17.40	335.00	249.00
12/23 ♂	12.70	6.12	111.00	0.32	52.30	18.10	347.00	238.00
13/23 ♀	10.30	5.5	(−) 97.00	(−) 0.29	52.70	17.60	334.00	434.00
14/23 ♂	(−) 9.90	6.25	(−) 107.00	0.32	51.20	17.10	334.00	413.00

## Data Availability

For further information, please contact the corresponding author.

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
