# Peer review of "Resorbable Patient-Specific Implants of Molybdenum for Pediatric Craniofacial Surgery—Proof of Concept in an In Vivo Pilot Study"

_jfb, 2024, doi:10.3390/jfb15050118_

Round 1

Reviewer 1 Report

Comments and Suggestions for Authors

I believe that the work being reviewed is valid from a scientific point of view and well written, although preliminary it deals with a topic of interest and certainly innovative. I therefore believe it is valid to be published.

Reviewer 2 Report

Comments and Suggestions for Authors

Interesting work. 

While the "Introduction" highlights and refers to the authors' recent work on molybdenum, in terms of biocompatibility, stability and absorption, such can be further improved to the JFB reader via providing more information on resorption/degradation testing, and explaining the experimental timeline (previous rat model @ 12, 24 and 52 weeks versus present pig model @ 54 days). To rephrase, explain/clarify how and why such euthanasia timepoints were selected. Herein, Figure 2 can be further improved, visually.

Further, Subsection "2.1.2. Implants" is brief and can be improved by providing more in-depth details, including listing of equipment used in the processes.

Subsection "2.2.8. Analysis of the Metallographic Cross Sections" can be further improved to provide more information to the non-expert reader.

Why not display more histopathology and only provide 2? 

Whilst the manuscript concludes no satisfactory resorption at 54 days, the authors state that further in vivo testing seem necessary, over a longer period of time .. henceforth connected to the previous inquiry on how/why this timeline was opted for. Finally, such is only limited to the in situ residency time or also can be attributed to the material design/properties?  

Reviewer 3 Report

Comments and Suggestions for Authors

The article concerns the implants used for oral and maxillofacial interventions, focusing mainly on the importance of developing absorbable implants, which therefore allow to avoid second removal operations.

Pre- and post- operative protocols are accurately described. I have no thorough knowledge for a critical correction of the medical part.

My comments:

- Please improve the resolution and contrast of Figures 5, 6 and 7.

- Please make comparisons with the expected values from titanium implants in reporting the values measured on animals in the post-operative period (blood analysis, evaluation of new bone formation, histopathological analysis, etc.)

- How was the duration selected? Since previous studies had already demonstrated the biocompatibility of the plant, the research did not provide further information on the reabsorption of molybdenum plants. Could a monitoring of the reabsorption of the plant with the CT scan have revealed the insufficient time selected? the removal of the implant only after obvious signs of resorption could have prevented the need to repeat the in vivo study with longer times, sacrificing other animals.

Reviewer 4 Report

Comments and Suggestions for Authors

1. The introduction contains too many unnecessary paragraphs. It is best to group the sentences appropriately and organize them into 3-4 paragraphs.

2. polylactic acids and polyglycolic acid (PLA) -> Are they grouped together and presented as an abbreviation? Since the two are completely different, it would be better to separate them and present them as abbreviations.

3. The CAD/CAM abbreviation was defined earlier, but the abbreviation is repeatedly defined again (P3 L128). Please correct it.

4. P4 126-138 will be described in Materials and Methods, and generally do not need to be described unnecessarily in the introduction. It is best to delete it, and at the end of the introduction, describe the purpose of this study.

5. 'the Niedersächsische Landesamt für Verbraucherschutz und Lebensmittelsicherheit' is translated as 'Lower Saxony State Office for Consumer Protection and Food Safety'. Was this animal study approved by The Institutional Animal Care and Use Committee (IACUC)? Comments here are important for animal experiment ethics and should not be omitted.

6. All figures in this manuscript are marked as (s. Figure XX). It is recommended to delete 's.' as it may be mistaken for supplementary.

7. What is GV Solas? Full name notation is required, and a corresponding reference appears to be required.

8. There is no mention of region of interest (ROI) for histologic evaluation. Is it simply bone tissue underneath the area where the implants were? A clear definition of ROI is necessary for readers to understand.

9. This study has a fatal flaw in its design. There is only a test group using 4 pigs. A scientific comparison cannot be made because there is no control group in which nothing was added or another group in which implants made of existing titanium or new Mg were applied. This should be clearly stated as a limitation of this study in the discussion, and the title also needs to be changed to in vivo pilot study.

10. As the authors mentioned, compared to other studies, the F/U period is short at 54 days. So, why was this experiment conducted with an F/U period of 54 days? Isn’t there a need to at least follow the period adopted in other similar studies?

11. As stated in the title, molybdenum implant for pediatric craniofacial surgery, it is generally necessary to mention how long it is ideal to be absorbed or removed when applying titanium or resorbable polymer implants in pediatric craniofacial surgery, and this study It seems necessary to mention this in conjunction with the F/U period of 54 days.

Comments on the Quality of English Language

Minor editing of English language is required
